# OPTIMIZATION FRAMEWORK OF TRANSFER LEARNING AND ITS FEASIBILITY

## ABSTRACT

Transfer learning is a fast developing paradigm for utilizing existing knowledge from previous learning tasks to improve the performance of new ones. It has enjoyed numerous empirical successes and inspired a growing number of theoretical studies. This paper addresses the feasibility issue of transfer learning. It begins by establishing the necessary mathematical concepts and constructing a mathematical framework for transfer learning. It then identifies and formulates the three-step transfer learning procedure as an optimization problem, allowing for the resolution of the feasibility issue. Importantly, it demonstrates that under certain technical conditions, such as appropriate choice of loss functions and data sets, an optimal procedure for transfer learning exists. This study of the feasibility issue brings additional insights into various transfer learning problems. It sheds light on the impact of feature augmentation on model performance, explores potential extensions of domain adaptation, and examines the feasibility of efficient feature extractor transfer in image classification.

## 1 INTRODUCTION

Transfer learning is a popular paradigm in machine learning. The basic idea of transfer learning is simple: it is to leverage knowledge from a well-studied learning problem, known as the source task, to enhance the performance of a new learning problem with similar features, known as the target task. In deep learning applications with limited and relevant data, it has become standard practice to employ transfer learning by utilizing large datasets (e.g., ImageNet) and their corresponding pre-trained models (e.g., ResNet50). Transfer learning has demonstrated success across various fields, including natural language processing (Ruder et al., 2019; Devlin et al., 2019; Sung et al., 2022), sentiment analysis (Jiang & Zhai, 2007; Deng et al., 2013; Liu et al., 2019), computer vision (Deng et al., 2009; Long et al., 2015; Ganin et al., 2016; Wang & Deng, 2018), activity recognition Cook et al. (2013); Wang et al. (2018), medical data analysis (Zeng et al., 2019; Wang et al., 2022; Kim et al., 2022), bio-informatics (Hwang & Kuang, 2010), finance (Leal et al., 2020; Rosenbaum & Zhang, 2021), recommendation system (Pan et al., 2010; Yuan et al., 2019), and fraud detection (Lebichot et al., 2020). (For further insights, refer to various review papers such as Pan & Yang (2010); Tan et al. (2018); Zhuang et al. (2020)). Transfer learning remains a versatile and enduring paradigm in the rapidly evolving AI landscape, where new machine learning techniques and tools emerge at a rapid pace.

Given the empirical successes of transfer learning, there is a growing body of theoretical work focused on transfer learning, particularly transferability. For instance, transferability in the domain adaptation setting is often quantified by measuring the similarity between the source and target domains using various divergences, including low-rank common information in Saenko et al. (2010), KL-divergence in Ganin & Lempitsky (2015); Ganin et al. (2016); Tzeng et al. (2017), $l_2$-distance in Long et al. (2014), the optimal transportation cost in Courty et al. (2017), and the Renyi divergence in Azizzadenesheli et al. (2019).

In classification tasks within the fine-tuning framework, transferability metrics and generalization bounds are derived under different measurements, such as the VC-dimension of the hypothesis space adopted in Blitzer et al. (2007), total variation distance in Ben-David et al. (2010), $f$-divergence in Harremoës & Vajda (2011), Jensen-Shannon divergence in Zhao et al. (2019), $\mathcal{H}$-score in Bao et al. (2019), negative conditional entropy between labels in Tran et al. (2019), mutual information in Bu

et al. (2020), $\mathcal{X}^2$-divergence in Tong et al. (2021), Bhattacharyya class separability in Pándy et al. (2022), and variations of optimal transport cost in Tan et al. (2021).

Recent research has aimed to design transferability metrics that encompass more general supervised learning tasks and deep learning models. For example, Mousavi Kalan et al. (2020) studied transfer learning with shallow layer neural networks and established the minimax generalization bound; Nguyen et al. (2020) measured transferability by computing the negative cross-entropy of soft labels generated by the pre-trained model. You et al. (2021) estimated transferability using the marginalized likelihood of labeled target data, assuming the addition of a linear classifier on top of the pre-trained deep learning model. Huang et al. (2022) introduced TransRate, a computationally-efficient and optimization-free transferability measure. Nguyen et al. (2022) bounded the transfer accuracy of a deep learning model using a quantity called the majority predictor accuracy. Additionally, theoretical bounds for transfer learning in the context of representation learning Tripuraneni et al. (2020) and few-shot learning Galanti et al. (2022) have also been explored.

Given the advancements made in both empirical and theoretical aspects of transfer learning, it is imperative that we address another fundamental issue: the feasibility of transfer learning.

Understanding the feasibility of transfer learning helps make informed decisions about when and how to apply transfer learning techniques. It also guides the development of appropriate algorithms, methodologies, and frameworks for effective knowledge transfer. By establishing the feasibility of transfer learning, we can unlock its potential for enhancing model performance, accelerating learning processes, and addressing data limitations in various real-world applications.

**Our work.** This paper addresses the feasibility issue of transfer learning through several steps. It begins by establishing the necessary mathematical concepts, and then constructs a comprehensive mathematical framework. This framework encompasses the general procedure of transfer learning by identifying its three key steps and components. Next, it formulates the three-step transfer learning procedure as an optimization problem, allowing for the resolution of the feasibility issue. Importantly, it demonstrates that under appropriate technical conditions, such as the choice of proper loss functions and compact data sets, an optimal procedure for transfer learning exists.

Furthermore, this study of the feasibility issue brings additional insights into various transfer learning problems. It sheds light on the impact of feature augmentation on model performance, explores potential extensions of domain adaptation, and examines the feasibility of efficient feature extractor transfer in the context of image classification.

## 2 MATHEMATICAL FRAMEWORK OF TRANSFER LEARNING

In this section, we will introduce necessary concepts and establish a mathematical framework for the entire procedure of transfer learning. For ease of exposition and without loss of generality, we will primarily focus on a supervised setting involving a source task $S$ and a target task $T$ on a probability space $(\Omega, \mathcal{F}, \mathbb{P})$.

To motivate the mathematical concepts and framework, we begin by revisiting some transfer problems.

### 2.1 EXAMPLES OF TRANSFER LEARNING

**Domain adaption.** This particular class of transfer learning problems is also known as *covariate shift* Saenko et al. (2010); Long et al. (2014); Ganin & Lempitsky (2015); Ganin et al. (2016); Courty et al. (2017); Tzeng et al. (2017); Azizzadenesheli et al. (2019). In domain adaptation, the crucial assumption is that the relation between input and output remain the same for both the source and the target tasks. As a result, the focus is to capture the difference between source and target inputs. Mathematically, this assumption implies that once the conditional distribution of the output variable given the input variable is learned from the source task, it suffices to derive an appropriate *input transport mapping* that aligns the distribution of the target inputs with that of the source inputs. This perspective, often referred to as the "optimal transport" view of transfer learning, has been extensively studied by Flamary et al. Courty et al. (2017).

**Image classification.**    This popular class of problems in transfer learning Tran et al. (2019); Bao et al. (2019); Tan et al. (2021); You et al. (2021); Huang et al. (2022) is typically addressed using a neural network approach. In this approach, the neural network structure comprises a *feature extractor* module, followed by a final *classifier layer*. Relevant studies, such as Bao et al. (2019) and Tan et al. (2021), often adopt this architecture. In this setup, only the last few layers of the model are retrained when solving the target task, while the feature extraction layers derived from the source task are directly utilized. This approach allows for leveraging the learned representations from the source task, optimizing the model specifically for the target task.

**Large language model.**    This class of problem such as Devlin et al. (2019); Xia et al. (2022) serves as a prominent testing ground for transfer learning techniques due to the scale of network models and the complexity of the data involved. A widely used example is the BERT model Devlin et al. (2019), which typically consists of neural networks with a substantial number of parameters, hence it usually starts with pretraining the model over a large and generic dataset, followed by a fine-tuning process for specific downstream tasks. Here, the pretraining process over generic datasets can be viewed as solving for the source task, and the designated downstream tasks can be categorized as target tasks. For instance, Xia et al. (2022) suggests a particular fining-tuning technique to better solve the target tasks. This technique combines *structure pruning with distillation*: after pretraining a large language model with multi-head self-attention layers and feed-forward layers, the study suggests applying a structure pruning technique to each layer. This pruning process selects a simplified sub-model specifically tailored for the designated downstream task. Subsequently, a distillation procedure ensures the transfer of most relevant knowledge to the pruned sub-model.

## 2.2   Mathematical framework for transfer learning

Built on the intuition of the previous transfer learning problems, we will now establish the rigorous mathematical framework of transfer learning, staring with fixing the notation for the source and the target tasks.

### 2.2.1   Source and target tasks in transfer learning

**Target task $T$.**    In the target task $T$, we denote $\mathcal{X}_T$ and $\mathcal{Y}_T$ as its input and output spaces, respectively, and $(X_T, Y_T)$ as a pair of $\mathcal{X}_T \times \mathcal{Y}_T$-valued random variables. Here, $(\mathcal{X}_T, \| \cdot \|_{\mathcal{X}_T})$ and $(\mathcal{Y}_T, \| \cdot \|_{\mathcal{Y}_T})$ are Banach spaces with norms $\| \cdot \|_{\mathcal{X}_T}$ and $\| \cdot \|_{\mathcal{Y}_T}$, respectively. Let $L_T : \mathcal{Y}_T \times \mathcal{Y}_T \to \mathbb{R}$ be a real-valued function, and assume that the learning objective for the target task is

$$\min_{f \in A_T} \mathcal{L}_T(f_T) = \min_{f_T \in A_T} \mathbb{E}[L_T(Y_T, f_T(X_T))], \tag{1}$$

where $\mathcal{L}_T(f_T)$ is a loss function that measures a model $f_T : \mathcal{X}_T \to \mathcal{Y}_T$ for the target task $T$, and $A_T$ denotes the set of target models such that

$$A_T \subset \{f_T | f_T : \mathcal{X}_T \to \mathcal{Y}_T\}. \tag{2}$$

Take the image classification task as an example, $\mathcal{X}_T$ is a space containing images as high dimensional vectors, $\mathcal{Y}_T$ is a space containing image labels, $(X_T, Y_T)$ is a pair of random variables satisfying the empirical distribution of target images and their corresponding labels, and $L_T$ is the cross-entropy loss function between the actual label $Y_T$ and the predicted label $f_T(X_T)$. For the image classification task using neural networks, $A_T$ will depend on the neural network architecture as well as the constraints applied to the network parameters.

Let $f_T^*$ denote the optimizer for the optimization problem equation 1, and $\mathbb{P}_T = Law(f_T^*(X_T))$ for the probability distribution of its output. Then the model distribution $\mathbb{P}_T$ depends on three factors: $L_T$, the conditional distribution $Law(Y_T|X_T)$, and the marginal distribution $Law(X_T)$. Note that in direct learning, this optimizer $f_T^* \in A_T$ is solved directly by analyzing the optimization problem equation 1, whereas in transfer learning, one leverages knowledge from the source task to facilitate the search of $f_T^*$.

**Source task $S$.**    In the source task $S$, we denote $\mathcal{X}_S$ and $\mathcal{Y}_S$ as the input and output spaces of the source task, respectively, and $(X_S, Y_S)$ as a pair of $\mathcal{X}_S \times \mathcal{Y}_S$-valued random variables. Here, $(\mathcal{X}_S, \| \cdot \|_{\mathcal{X}_S})$ and $(\mathcal{Y}_S, \| \cdot \|_{\mathcal{Y}_S})$ are Banach spaces with norms $\| \cdot \|_{\mathcal{X}_S}$ and $\| \cdot \|_{\mathcal{Y}_S}$, respectively. Let

$L_S : \mathcal{Y}_S \times \mathcal{Y}_S \to \mathbb{R}$ be a real-valued function and let us assume that the learning objective for the source task is

$$\min_{f_S \in A_S} \mathcal{L}_S(f_S) = \min_{f \in A_S} \mathbb{E}[L_S(Y_S, f_S(X_S))], \tag{3}$$

where $\mathcal{L}_S(f_S)$ is the loss function for a model $f_S : \mathcal{X}_S \to \mathcal{Y}_S$ for the source task $S$. Here $A_S$ denotes the set of source task models such that

$$A_S \subset \{f_S | f_S : \mathcal{X}_S \to \mathcal{Y}_S\}. \tag{4}$$

Moreover, denote the optimal solution for this optimization problem equation 3 as $f_S^*$, and the probability distribution of the output of $f_S^*$ by $\mathbb{P}_S = Law(f_S^*(X_S))$. Meanwhile, similar as the target model, the model distribution $\mathbb{P}_S$ will depend on the function $L_S$, the conditional distribution $Law(Y_S|X_S)$, and the marginal distribution $Law(X_S)$.

Back to the image classification example, the target task may only contain images of items in an office environment, the source task may have more image samples from a richer dataset, e.g., ImageNet. Meanwhile, $\mathcal{X}_S$ and $\mathcal{Y}_S$ may have different dimensions compared with $\mathcal{X}_T$ and $\mathcal{Y}_T$, since the image resolution and the class number vary from task to task. Similar to the admissible set $A_T$ in the target task, $A_S$ depends on the task description, and $f_S^*$ is usually a deep neural network with parameters pretrained using the source data.

In transfer learning, the optimal model $f_S^*$ for the source task is also referred to as a pretrained model. The essence of transfer learning is to utilize this pretrained model $f_S^*$ from the source task to accomplish the optimization objective equation 1. We now define this procedure in three steps.

### 2.2.2 THREE-STEP TRANSFER LEARNING PROCEDURE

**Step 1. Input transport.** Since $\mathcal{X}_T$ is not necessarily contained by the source input space $\mathcal{X}_S$, the first step is therefore to make an appropriate adaptation to the target input $X_T \in \mathcal{X}_T$. In the example of image classification, popular choices for input transport may include resizing, cropping, rotation, and grayscale. We define this adaptation as an *input transport mapping*.

**Definition 1** (Input transport mapping). *A function*

$$T^X \in \{f_{input} | f_{input} : \mathcal{X}_T \to \mathcal{X}_S\} \tag{5}$$

*is called an input transport mapping with respect to the source and target task pair $(S, T)$ if it takes any data point in the target input space $\mathcal{X}_T$ and maps it into the source input space $\mathcal{X}_S$.*

With an input transport mapping $T^X$, the first step of transfer learning can be represented as follows.

$$\mathcal{X}_T \ni X_T \xmapsto{\text{Step 1. Input transport by } T^X} T^X(X_T) \in \mathcal{X}_S.$$

Recall that in domain adaption, it is assumed that the difference between the source input distribution $Law(X_S)$ and target input distribution $Law(X_T)$ is the only factor to motivate the transfer. Therefore, once a proper input transport mapping $T^X$ is found, transfer learning is accomplished. Definition 1 is thus consistent with Courty et al. (2017), in which domain adaption is formulated as an optimal transport from the target input to the source input.

For most transfer learning problems, however, one needs both a transport mapping for the input *and* a transport mapping for the output. For instance, the labeling function for different classes of computer vision tasks, such as object detection, instance segmentation, and image classification, can vary greatly and depend on the specific task. Hence, the following two more steps are required.

**Step 2. Applying pretrained model.** After applying an input transport mapping $T^X$ to the target input $X_T$, the pretrained model $f_S^*$ will take the transported data $T^X(X_T) \in \mathcal{X}_S$ as an input. That is,

$$\mathcal{X}_S \ni T^X(X_T) \xmapsto{\text{Step 2. Apply } f_S^*} (f_S^* \circ T^X)(X_T) \in \mathcal{Y}_S,$$

where $(f_S^* \circ T^X)(X_T)$ denotes the corresponding output of the pretrained model $f_S^*$. Note here the composed function $f_S^* \circ T^X \in \{f_{\text{int}} | f_{\text{int}} : \mathcal{X}_T \to \mathcal{Y}_S\}$.

**Step 3. Output transport.** After utilizing the pretrained model $f_S^*$, the resulting model $f_S^* \circ T^X \in \{f_{\text{int}}|f_{\text{int}} : \mathcal{X}_T \to \mathcal{Y}_S\}$ may still be inadequate for the target model: one may need to map the $\mathcal{Y}_S$-valued output into the target output space $\mathcal{Y}_T$ and in many cases such as image classification or large language models, $\mathcal{Y}_S$ and $\mathcal{Y}_T$ do not necessarily coincide. Besides, more fine-tuning steps are needed for problems other than domain adaptation. Hence, it is necessary to define an *output transport mapping* to map an intermediate model from $\{f_{\text{int}}|f_{\text{int}} : \mathcal{X}_T \to \mathcal{Y}_S\}$ to a target model in $A_T$.

**Definition 2** (Output transport mapping). *A function*

$$T^Y \in \{f_{output}|f_{output} : \mathcal{X}_T \times \mathcal{Y}_S \to \mathcal{Y}_T\} \tag{6}$$

*is called an output transport mapping with respect to the source and target task pair $(S, T)$ if, for an optimal source model $f_S^* : \mathcal{X}_S \to \mathcal{Y}_S$ and an input transport mapping $T^X$ as in Definition 1, the composed function $T^Y(\cdot, f_S^* \circ T^X(\cdot)) \in A_T$.*

This output transport mapping can be further tailored to adapt to more complex models; see, for instance, the discussion of large language models in Section 2.3. Many popular applications of transfer learning contain an output mapping component as in Definition 2. Take the aforementioned image classification in Section 2.1: after adopting the feature extractor $f_S^*$ obtained from the source task, an additional classifier layer is attached after the module of $f_S^*$ in the network structure and will be fine-tuned for the target task. This classifier layer takes the exact role of the output transport mapping.

Now, this third and the final step in transfer learning can be expressed as

$$\mathcal{X}_T \times \mathcal{Y}_S \ni (X_T, (f_S^* \circ T^X)(X_T)) \xmapsto{\text{Step 3. Output transport by } T^Y} T^Y\left(X_T, (f_S^* \circ T^X)(X_T)\right) \in \mathcal{Y}_T.$$

Combining these three steps, transfer learning can be presented by the following diagram,

$$
\begin{array}{ccc}
\mathcal{X}_S \ni X_S & \xRightarrow{\text{Pretrained model } f_S^* \text{ from } equation\ 3} & f_S^*(X_S) \in \mathcal{Y}_S \\
T^X \Uparrow & & \Downarrow T^Y \\
\mathcal{X}_T \ni X_T & \xdashrightarrow[f_T^* \in \arg\min_{f \in A_T} \mathcal{L}_T(f_T)]{\text{Direct learning equation 1}} & f_T^*(X_T) \in \mathcal{Y}_T
\end{array}
\tag{7}
$$

In summary, transfer learning aims to find an appropriate pair of input and output transport mappings $T^X$ and $T^Y$, where the input transport mapping $T^X$ translates the target input $X_T$ back to the source input space $\mathcal{X}_S$ in order to utilize the optimal source model $f_S^*$, and the output transport mapping $T^Y$ transforms a $\mathcal{Y}_S$-valued model to a $\mathcal{Y}_T$-valued model. This is in contrast to the direct learning, where the optimal model $f_T^*$ is derived by solving the optimization problem in the target task equation 1. In other words, transfer learning is the following optimization problem.

**Definition 3** (Transfer learning). *The three-step transfer learning procedure presented in equation 7 is to solve the optimization problem*

$$\min_{T^X \in \mathbb{T}^X, T^Y \in \mathbb{T}^Y} \mathcal{L}_T\left(T^Y(\cdot, (f_S^* \circ T^X)(\cdot))\right) := \min_{T^X \in \mathbb{T}^X, T^Y \in \mathbb{T}^Y} \mathbb{E}\left[L_T\left(Y_T, T^Y(X_T, (f_S^* \circ T^X)(X_T))\right)\right].$$
$$\tag{8}$$

*Here, $\mathbb{T}^X$ and $\mathbb{T}^Y$ are proper sets of transport mappings such that*

$$\{T^Y(\cdot, (f_S^* \circ T^X)(\cdot))|T^X \in \mathbb{T}^X, T^Y \in \mathbb{T}^Y\} \subset A_T.$$

*In particular, when $\mathcal{X}_S = \mathcal{X}_T$ (resp. $\mathcal{Y}_S = \mathcal{Y}_T$), the identity mapping $id^X(x) = x$ (resp. $id^Y(x, y) = y$) is included in $\mathbb{T}^X$ (resp. $\mathbb{T}^Y$).*

Let us reexamine the aforementioned examples of transfer learning, from this new optimization perspective.

## 2.3 EXAMPLES OF TRANSFER LEARNING THROUGH THE LENS OF OPTIMIZATION

**Domain adaption.** Here we define the family of admissible output transport mappings as $\mathbb{T}^Y = \{\text{id}_\mathcal{Y}\}$, where $\text{id}_\mathcal{Y}$ denotes the identity mapping on $\mathcal{Y}$; define the family of admissible input transport

mappings as $\mathbb{T}^X = \{T^X : \mathcal{X}_T \to \mathcal{X}_S \,|\, T^X \text{ is one-to-one}\}$. When the output variables for both the source and the target tasks coincide such that $Y_S = Y_Y = Y$, and when the loss functions for both tasks take the same form such that $L_S = L_T = L : \mathcal{Y} \times \mathcal{Y} \to \mathbb{R}$, then $T^{*X}$ is the optimal solution to the optimization problem equation 8 taking a particular form of

$$\min_{T^X \in \mathbb{T}^X} \mathbb{E}[L(Y, f_S^*(T^X(X_T)))]. \tag{9}$$

Moreover, it can be shown that the optimal source model and optimal target model satisfy the relation $f_T^* = f_S^* \circ T^{X,*}$, where

$$f_S^* := \underset{f_S:\mathcal{X}_S \to \mathcal{Y}}{\arg\min} \mathbb{E}[L(Y, f_S(X_S))], \quad f_T^* := \underset{f_T:\mathcal{X}_T \to \mathcal{Y}}{\arg\min} \mathbb{E}[L(Y, f_T(X_T))].$$

That is, solving the transfer learning problem is reduced to finding an optimal input transport mapping $T^{X,*}$, given the pre-trained model $f_S^*$. This is exactly domain adaptation.

**Image classification.** For this class of problems, we take the transfer learning problem over a benchmark dataset, the Office-31 Saenko et al. (2010), as an example. This dataset consists of images from three domains: Amazon (A), Webcam (W), and DSLR (D), containing 4110 images of 31 categories of objects in an office environment.

Here, the source task $S$ can be chosen from any of three domains (A, D, or W), where all input images are first resized into dimension $3 \times 244 \times 244$, that is, $\mathcal{X}_S \subset \mathbb{R}^{3 \times 244 \times 244}$ being the space of resized image samples from the source domain, and

$$\overline{\mathcal{Y}}_S = \Delta_{31} := \{p \in \mathbb{R}^{31} : \sum_1^{31} p_i = 1, p_i \geq 0, \forall 1 \leq i \leq 31\}$$

being the space of image class labels. Since the purpose of solving this source task is to derive the feature extractor module implemented as a ResNet50 network structure in Figure 1, we define the effective source output space as the feature space, $\mathcal{Y}_S \subset \mathbb{R}^{2048}$. For any target task $T$ (A, D, or W) different from that of $S$,

$$\mathcal{X}_T = \mathcal{X}_S \subset \mathbb{R}^{3 \times 244 \times 244}$$

is the space of resized image samples from the target domain, and the output space is set to be $\mathcal{Y}_T = \overline{\mathcal{Y}}_S = \Delta_{31}$. For both the source and the target tasks, the loss function $L_S = L_T$ is chosen to be the cross entropy between the actual label and the predicted label.

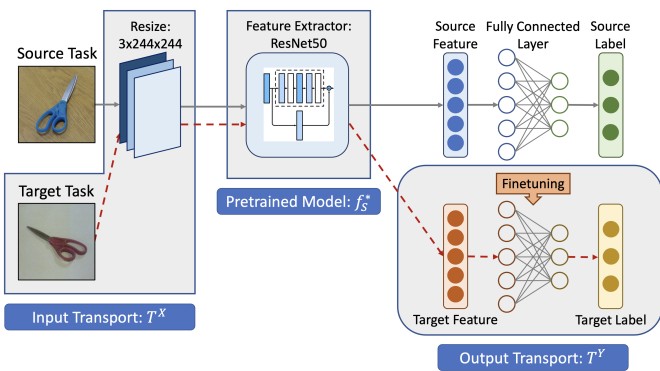

Figure 1: Illustration of input transport $T^X$, pretrained model $f_S^*$ and output transport $T^Y$ in the Office-31 transfer learning task.

As introduced in Figure 1, the set of source models are given by

$$A_S = \{f_{\text{NN}} \circ f_{\text{Res}} : \mathcal{X}_S \to \mathcal{Y}_S | f_{\text{NN}} \in \text{NN}_{2048}^{31}, f_{\text{Res}} \in \text{Res}_{3 \times 244 \times 244}^{2048}\}.$$

Here $\text{Res}_{3 \times 244 \times 244}^{2048}$ denotes all ResNet50 architectures with $3 \times 244 \times 244$-dimensional input and 2048-dimensional output, and $\text{NN}_{2048}^{31}$ denotes all two-layer neural networks which map a 2048-dimensional feature vector to a 31-dimensional probability vector in $\mathcal{Y}_S$. The source model $f_{\text{Res},S}^*$ and $f_{\text{NN},S}^*$ is obtained by solving the source task optimization equation 3.

To transfer the source task to the target task, the pretrained ResNet50 model $f_{\text{Res},S}^*$ will be fixed, while the last two-layer classifier $f_{\text{NN}} \in \text{NN}_{2048}^{31}$ will be fine-tuned using part of the data from the target domain $(\mathcal{X}_T, \mathcal{Y}_T)$. In this case, the input transport set $\mathbb{T}^X$ is a singleton set whose element is the identity mapping on $\mathbb{R}^{3 \times 244 \times 244}$, while the output transport mapping $T^Y$ is a two-layer classifier from the corresponding set $\mathbb{T}^Y$ given by

$$\mathbb{T}^Y = \{f_{\text{NN}} | f_{\text{NN}} \in \text{NN}_{2048}^{31}\}. \tag{10}$$

Meanwhile, the set of admissible target models is given by

$$A_T = \{f_{\text{NN}} \circ f_{\text{Res},S}^* : \mathcal{X}_T \to \mathcal{Y}_T | f_{\text{NN}} \in \text{NN}_{2048}^{31}\}, \tag{11}$$

and the transfer learning task is formulated as $\min_{T^Y \in \mathbb{T}^Y} \mathbb{E}\left[L_T\left(Y_T, T^Y(X_T)\right)\right]$. Note the formulation is slightly simpler than equation 8 because in this particular example, the output transport in $\mathbb{T}^Y$ takes inputs from $\mathcal{X}_T$ instead of $\mathcal{X}_T \times \mathcal{Y}_S$.

**Large language models.** Following the discussion in Section 2.1 on the large language models such as in Xia et al. (2022), the combined operation of structure pruning and distillation can be interpreted as an extended form of output transport mapping: it is an operator

$$T^Y : \{f_{\text{int}} | f_{\text{int}} : \mathcal{X}_T \to \mathcal{Y}_S\} \to \{f_T | f_T : \mathcal{X}_T \to \mathcal{Y}_T\} \tag{12}$$

such that for an optimal source model $f_S^* : \mathcal{X}_S \to \mathcal{Y}_S$ and an input transport mapping $T^X$ as in Definition 1, the output $T^Y(f_S^* \circ T^X) \in A_T$. In these models, combining structure pruning and distillation technique is shown to improve the performance of the pretrained model $f_S^*$: pruning eliminates unnecessary parameters in the pretrained model, and the distillation filters out irrelevant information with proper adjustment of model parameters. From Xia et al. (2022) we observe that the design of the output transport mapping $T^Y$ depends on the target input data and is tailored to the specific input dataset.

## 3 Feasibility of Transfer Learning as an Optimization Problem

The above optimization reformulation of the three-step transfer learning procedure provides a unified framework to analyze the impact and implications of various transfer learning techniques. In particulr, it enables analyzing the feasibility of transfer learning. We show that under appropriate technical conditions, there exists an optimal procedure for transfer learning, i.e., the pair of transport mappings $(T^{X,*}, T^{Y,*})$ for equation 8.

### 3.1 Feasibility of Transfer Learning

To facilitate the feasibility analysis, the following class of loss function $\mathcal{L}_T$ is introduced.

**Definition 4** (Proper loss function). *Let $(X, Y)$ be a pair of $\mathcal{X}_T \times \mathcal{Y}_T$-valued random variables with $Law(X_T, Y_T) \in \mathcal{P}(\mathcal{X}_T \times \mathcal{Y}_T)$. A loss functional $\mathcal{L}_T$ over $A_T$ is said to be* proper *with respect to $(X, Y)$ if there exist a corresponding function $L_T : \mathcal{Y}_T \times \mathcal{Y}_T \to \mathbb{R}$ bounded from below such that for any $f \in A_T$, $\mathcal{L}_T(f) = \mathbb{E}[L_T(Y, f(X))] = \mathbb{E}[\mathbb{E}[L_T(Y, f(X))|X]]$; moreover, the function $\tilde{L}_T : \mathcal{Y}_T \to \mathbb{R}$, given by $\tilde{L}_T(y) = \mathbb{E}[L_T(Y, Y')|Y' = y]$ for all $y \in \mathcal{Y}_T$, is continuous.*

Examples of proper loss functions include mean squared error and KL-divergence and more generally the Bregman divergence Banerjee et al. (2005) given by

$$D_\phi(u, v) = \phi(u) - \phi(v) - \langle u - v, \nabla\phi(v) \rangle \tag{13}$$

for some strictly convex and differentiable $\phi : \mathcal{Y} \to \mathbb{R}$, assuming that the first and second moments of $Y$ conditioned on $Y' = y$ is continuous with respect to $y$.

Without loss of generality, we shall in this section assume the input transport set $\mathbb{T}^X$ contains all functions from $\mathcal{X}_T$ to $\mathcal{X}_S$. We then specify the following assumptions for the well-definedness of equation 8.

**Assumption 1.** *1. $\mathcal{L}_T$ is a proper loss functional with respect to $(X_T, Y_T)$;*

*2. the image $f_S^*(\mathcal{X}_S)$ is compact in $(\mathcal{Y}_S, \|\cdot\|_{\mathcal{Y}_S})$;*

3. *the set $\mathbb{T}^Y \subset \mathcal{C}(\mathcal{X}_T; \mathcal{Y}_T)$ is such that the following set of functions*

$$\tilde{\mathbb{T}}^Y = \{\tilde{T}^Y : \mathcal{X}_T \to \mathcal{Y}_T \mid \exists T^Y \in \mathbb{T}^Y \text{ s.t. } \tilde{L}_T(\tilde{T}^Y(x)) = \inf_{y \in f_S^*(\mathcal{X}_S)} \tilde{L}_T(T^Y(x,y)), \; \forall x \in \mathcal{X}_T\}$$

*is compact in $(\{f|f : \mathcal{X}_T \to \mathcal{Y}_T\}, \|\cdot\|_\infty)$, where for any $f : \mathcal{X}_T \to \mathcal{Y}_T$, $\|f\|_\infty :=$ $\sup_{x \in \mathcal{X}_T} \|f(x)\|_{\mathcal{Y}_T}$.*

Popular choices of loss functions, such as mean squared error from the Bregman loss family, are not only proper but also strongly convex, therefore the compactness assumptions can be removed. Otherwise, compactness condition can be implemented by choosing a particular family of activation functions or imposing boundaries restrictions to weights and biases when constructing machine learning models.

Now we are ready to establish the following feasibility result.

**Theorem 1.** *There exists an optimal solution $(T^{X,*}, T^{Y,*}) \in \mathbb{T}^X \times \mathbb{T}^Y$ for optimization problem equation 8 under Assumption 1.*

*Proof of Theorem 1.* Since $\mathcal{L}_T$ is proper, there exists a function $L_T : \mathcal{Y}_T \times \mathcal{Y}_T \to \mathbb{R}$ such that $\inf_{(y,y') \in \mathcal{Y}_T \times \mathcal{Y}_T} L_T(y, y') > -\infty$, and $\mathcal{L}_T(T^Y(\cdot, (f_S^* \circ T^X)(\cdot))) = \mathbb{E}[L_T(Y_T, T^Y(X_T, (f_S^* \circ T^X)(X_T)))]$ for all $T^X \in \mathbb{T}^X$, $T^X \in \mathbb{T}^X$. Therefore, for the function $\tilde{L}_T(\cdot) = \mathbb{E}[L_T(Y, Y')|Y' = \cdot]$, there exists $m \in \mathbb{R}$ such that $\tilde{L}_T(y) \geq m$ for any $y \in \mathcal{Y}_T$.

Now fix any $T^Y \in \mathbb{T}^Y$. The continuity of $\tilde{L}_T$ and the continuity of $T^Y(x, \cdot)$ for each $x \in \mathcal{X}_T$ guarantee the continuity of $\tilde{L}_T(T^y(x, \cdot))$. Together with the compactness of $f_S^*(\mathcal{X}_S)$, we have that for any $x \in \mathcal{X}_T$,

$$M_{T^Y}^x := \arg\min_{y \in f_S^*(\mathcal{X}_S)} \tilde{L}_T(T^Y(x,y)) \neq \emptyset. \tag{14}$$

Therefore, for any $T^Y \in \mathbb{T}^Y$ and its corresponding $\tilde{T}^Y \in \tilde{\mathbb{T}}^Y$, one can construct $\tilde{T}^X \in \mathbb{T}^X$ such that $f_S^*(\tilde{T}^X(x)) \in M_{T^Y}^x$ for any $x \in \mathcal{X}_T$ and hence we have

$$\min_{T^X \in \mathbb{T}^X} \mathcal{L}_T(T^Y(\cdot, (f_S^* \circ T^X)(\cdot))) = \mathbb{E}[\tilde{L}_T(\tilde{T}^Y(X_T))] =: \tilde{\mathcal{L}}_T(\tilde{T}^Y).$$

The continuity of the new loss functional $\tilde{\mathcal{L}}_T$ comes from the continuity of the function $\tilde{L}$, and the particular choice of the function space $(\{f|f : \mathcal{X}_T \to \mathcal{Y}_T\}, \|\cdot\|_\infty)$, where $\{f|f : \mathcal{X}_T \to \mathcal{Y}_T\}$ contains all functions from $\mathcal{X}_T$ to $\mathcal{Y}_T$. Since $\tilde{\mathbb{T}}^Y$ is compact in $(\{f|f : \mathcal{X}_T \to \mathcal{Y}_T\}, \|\cdot\|_\infty)$, the minimum over $\tilde{\mathbb{T}}^Y$ is attained at some $\tilde{T}^{Y,*}$. According to the definition of $\tilde{\mathbb{T}}^Y$, there exists $T^{Y,*} \in \mathbb{T}^Y$ such that $\tilde{\mathcal{L}}_T(\tilde{T}^{Y,*}(\cdot)) = \inf_{y \in f_S^*(\mathcal{X}_S)} \tilde{\mathcal{L}}_T T^{Y,*}(\cdot, y)$. Let $T^{X,*}$ be the $\tilde{T}^X \in \mathbb{T}^X$ corresponding to $T^{Y,*}$. For any $T^X \in \mathbb{T}^X$ and $T^Y \in \mathbb{T}^Y$, we have

$$\mathcal{L}_T(T^Y(\cdot, (f_S^* \circ T^X)(\cdot))) \geq \mathcal{L}_T(T^Y(\cdot, (f_S^* \circ \tilde{T}^X)(\cdot))) = \tilde{\mathcal{L}}_T(\tilde{T}^Y(\cdot)) \geq \tilde{\mathcal{L}}_T(\tilde{T}^{Y,*}(\cdot))$$
$$= \mathcal{L}_T(T^{Y,*}(\cdot, (f_S^* \circ T^{X,*}))(\cdot)) \geq \min_{T^X \in \mathbb{T}^X, T^Y \in \mathbb{T}^Y} \mathcal{L}_T\left(T^Y(\cdot, (f_S^* \circ T^X)(\cdot))\right).$$

Therefore, the transfer learning problem equation 8 is well-defined and it attains its minimum at $(T^{X,*}, T^{Y,*})$ described above. □

## 3.2 Discussion

We now demonstrate that the feasibility analysis puts existing transfer learning studies on a firm mathematical footing, including domain adaptation and image classification. Additionally, it provides valuable insight for feature augmentation in particular, and expands the potential for improving model performance in general.

**Feasibility of domain adaption.** Following the discussion on the domain adaption problem in Section 2.3, the feasibility of the transfer learning framework equation 8 is clearly guaranteed: this is attributed to the optimality of the pretrained model $f_S^*$ inherited from the source optimization problem and the existence of an optimal transport mapping $T^{X,*}$ from $Law(X_T)$ to $Law(X_S)$.

Furthermore, for transfer learning problems not satisfying the usual premise of domain adaption, our framework enables introducing an output transport mapping, which allows for the alignment of the output distributions between the source and target tasks.

**Feasibility of image classification.** Take the aforementioned classification problems in Section 2.3 as an example. In practice, cross-entropy loss is convex with respect to the predicted probability vector, and the sigmoid activation function for the classifier layer ensures the the compactness assumption on $\mathbb{T}^Y$. For image data, $\mathcal{X}_S$ is typically a compact subset of an Euclidean space and therefore the image set for a continuous ResNet50 network is compact in the feature space. Hence the feasibility result holds. Our feasibility analysis provides the flexibility of incorporating an input transport mapping: it is feasible, and in fact beneficial for effectively utilizing the transferred feature extractor as investigated in Wang et al. (2022).

**Feasibility with feature augmentation.** Feature augmentation refers to the process of expanding the set of features used in a machine learning problem, which plays a significant role in improving the performance and effectiveness of models Volpi et al. (2018); Chen et al. (2019); Li et al. (2021). Importantly, transfer learning combined with feature augmentation can be integrated into the mathematical framework presented in Definition 3, enabling the feasibility of feature augmentation to be established accordingly. Specifically, in transfer learning with feature augmentation, we consider a source task $S$ with input and output variables $X \in \mathcal{X}$ and $Y \in \mathcal{Y}$. The target task involves predicting the same output $Y$ from $X$ along with an additional feature denoted by $Z \in \mathcal{Z}$, with:

$$\text{Source task:} \quad \min_{f:\mathcal{X}\to\mathcal{Y}} \mathbb{E}\left[D_\phi(Y, f(X))\right], \quad \text{Target task:} \quad \min_{f:\mathcal{X}\times\mathcal{Z}\to\mathcal{Y}} \mathbb{E}\left[D_\phi(Y, f(X, Z))\right]. \quad (15)$$

According to the feasibility result in Theorem 1, the loss functions in equation 15 can be selected as the Bregman loss in equation 13.

Moreover, the following result shows that, under the special case of "redundant information", transfer learning with feature augmentation can be solve explicitly by finding the appropriate input and output transport mappings.

**Corollary 1.** *Assume $Y$ and $Z$ are independent conditioned on $X$. The optimal input and output transport mappings $(T^X, T^Y)$ in the transfer learning optimization problem equation 8 under the feature augmentation setting equation 15 is given by*

$$T^X(x, z) = Proj_\mathcal{X}(x, z) = x, \quad and \quad T^Y(y) = id_\mathcal{Y}(y) = y.$$

Moreover, we have

**Corollary 2.** *Let $(T^{X,*}, T^{Y,*})$ be the optimal input and output transport mappings from solving the transfer learning optimization problem equation 8 under the feature augmentation setting equation 15, i.e.,*

$$(T^{X,*}, T^{Y,*}) = \underset{T^X\in\mathbb{T}^X, T^Y\in\mathbb{T}^Y}{\arg\min} \mathbb{E}\left[D_\phi\left(Y, T^Y(X, Z, (f_S^* \circ T^X)(X, Z))\right)\right], \quad (16)$$

*where $f_S^* = \arg\min_{f:\mathcal{X}\to\mathcal{Y}} \mathbb{E}\left[D_\phi(Y, f(X))\right]$ is the optimal pretrained model. Then,*

$$\mathbb{E}\left[D_\phi\left(Y, T^{Y,*}(X, Z, (f_S^* \circ T^{X,*})(X, Z))\right)\right] \leq \mathbb{E}\left[D_\phi(Y, f_S^*(X))\right]. \quad (17)$$

*Proof of Corollary 1 and 2.* First recall that under the Bregman loss, the optimal source and target models in equation 15 are given by the conditional expectations $f_S^*(X) = \mathbb{E}[Y|X]$ and $f_T^*(X, Z) = \mathbb{E}[Y|X, Z]$ (see Banerjee et al. (2005) for more details). Then, Corollary 1 follows from the fact that when $Y$ and $Z$ are independent conditioned on $X$, $\mathbb{E}[Y|X] = \mathbb{E}[Y|X, Z]$. Moreover, notice that $Proj_\mathcal{X} \in \mathbb{T}^X$ and $id_\mathcal{Y} \in \mathbb{T}^Y$ and Corollary 2 follows from the optimality of $(T^{X,*}, T^{Y,*})$. □

Corollary 1 suggests that if the added feature $Z$ does not provide more relevant information compared to the original feature $X$, transfer learning can be accomplished by discarding the additional feature and directly applying the pretrained model. Moreover, Corollary 2 demonstrates that incorporating additional information in transfer learning will not have any negative impact on model performance. In other words, the inclusion of supplementary information through transfer learning can, at worst, maintain the same level of model performance, and in general, can lead to performance improvement.

## 4 CONCLUSION

This paper establishes a mathematical framework for transfer learning, and resolves its feasibility issue. This study opens up new avenues for enhancing model performance, expanding the scope of transfer learning applications, and improving the efficiency of transfer learning techniques.

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
