# OpenReview forum: "Optimization Framework of Transfer Learning and its Feasibility"
_ICLR.cc/2024/Conference — Submitted to ICLR 2024_

### Official Review · Reviewer_QBcv · 2023-10-24

**Soundness:** 3 good
**Presentation:** 3 good
**Contribution:** 2 fair
**Rating:** 3
**Confidence:** 3

**Summary:**

This paper studies a mathematical formulation of transfer learning. Firstly, a class of loss functions is defined. Then, the problem is formulated as the optimization problem of the expected loss function under the population distribution. Finally, the existence of the optimal solution is proved. Based on the proof of the existence theorem, some theoretical properties of the optimal solution are investigated. Finally, the authors reveal that the added feature independent of Y conditioned on X does not convey relevant information.

**Strengths:**

The authors rigorously proved the existence of the optimal solution for the optimization problem of transfer learning.

**Weaknesses:**

Most part of the paper is devoted to preparing mathematical descriptions of transfer learning. The authors revealed rigorous assumptions for the existence of the optimal solution. However, the insight clarified by the theorems is not very clear. From the mathematical viewpoint, proving the existence of the optimal solution seems to be the starting point of the argument for statistical learning. However, the impact of this work on the machine-learning community seems to be quite limited. Not only the existence of the optimal solution but also additional theoretical results that have a practical impact would be necessary as an ICLR paper.

In transfer learning, the "negative transfer" phenomenon is an important issue. The authors could pursue the relationship between the theoretical result in the paper and negative transfer.

**Questions:**

- The authors revealed rigorous assumptions for the existence of the optimal solution. However, the insight clarified by the theorems is not very clear. Can the authors add usefulness to their analysis from a practical standpoint?

- In transfer learning, the "negative transfer" phenomenon is an important issue. Is there any relationship between the discussion in Section 3.2 and the negative transfer? In other words, can we use the theoretical result in the paper to reduce the effect of the negative transfer?

- Recently, numerous works for self-supervised learning have been published to elucidate the success of the large language model (LLM). Is it possible to apply the theoretical result in the paper to reveal the reason why the LLM works so efficiently?

---

### Official Review · Reviewer_Uz8e · 2023-10-29

**Soundness:** 4 excellent
**Presentation:** 4 excellent
**Contribution:** 2 fair
**Rating:** 3
**Confidence:** 3

**Summary:**

This paper establishes the mathematical concepts necessary for transfer learning and builds a mathematical framework for transfer learning. To this end, this paper describes a three-step transfer learning procedure: 1) Input transport, 2) Applying pretrained model, and 3) Output transport.  Then, this paper analyzes the feasibility of transfer learning as an optimization problem.

**Strengths:**

This paper integrates the problems of transfer learning into one mathematical framework and deals with them as optimization problems. Then, for each problem, what previous studies have solved was mathematically specified.

**Weaknesses:**

1. This paper covered the examples of transfer learning but not one of the important transfer learning problems, inductive transfer learning.
2. It is excellent to establish a mathematical framework for transfer learning, but this is not new. This paper seems to be a review paper on transfer learning and does not seem suitable for the ICLR conference.

**Questions:**

1. What new insights does the framework presented in this paper give us that we haven't seen in other studies?
2. Can inductive transfer learning be dealt with and integrated into this framework?
3. Wouldn't it be better to refer to the input mapping function as heterogeneous transfer learning learning according to Pan & Yang's nomenclature?

---

### Official Review · Reviewer_yVrx · 2023-10-31

**Soundness:** 2 fair
**Presentation:** 2 fair
**Contribution:** 2 fair
**Rating:** 5
**Confidence:** 4

**Summary:**

This paper presents an optimized perspective on transfer learning, encompassing two pivotal functions that require optimization: the input transformation function and the output transformation function. It convincingly illustrates that, given certain suitable conditions, an optimal solution is indeed attainable.

**Strengths:**

- The comprehensive three-step procedure delineated for transfer learning encompasses all possible scenarios wherein leveraging the source model is beneficial for addressing the target task, ensuring alignment in both input and output dimensions.
- The optimization perspective of transfer learning presented here is intriguing and offers valuable insights.

**Weaknesses:**

- The writing could benefit from improvement. The explanation of certain critical concepts, such as the proper loss function, is not sufficiently clear. Additionally, the assumptions require further elaboration, especially the third assumption. How do these assumptions align with common scenarios? A more systematic discussion would be highly advantageous.
- The primary theorem in this paper demonstrates that under specific conditions, the optimal input mapping and output mapping do exist. This theorem provides a theoretical foundation for the proposed optimization procedure in transfer learning. However, it still poses challenges in guiding us towards algorithm design.
- While there are classical theorems in transfer learning and domain adaptation, such as H-divergence, a more detailed comparison with these theorems is lacking. Furthermore, the distinctions from other theoretical works have not been adequately elucidated.

**Questions:**

- It would be advantageous to explicitly address the distinctions between the proposed framework and the corresponding theorem in comparison to other theoretical analyses in the field of transfer learning.
- The third assumption requires more detailed explanation. Could you please provide further elaboration and discuss how easily it can be satisfied in common scenarios?
- In many cases, even if we are aware that optimal input mapping and output mapping exist, obtaining such mappings remains challenging. Often, we can only attain near-optimal solutions. How do we quantify the performance gap between the optimal solution and the one achievable in real-world scenarios within the proposed framework?

---

### Official Review · Reviewer_f6nM · 2023-11-02

**Soundness:** 3 good
**Presentation:** 3 good
**Contribution:** 2 fair
**Rating:** 3
**Confidence:** 4

**Summary:**

This paper establishes a mathematical framework for transfer learning, and resolves its feasibility issue. It then identifies and formulates the three-step transfer learning procedure as an optimization problem, allowing for the resolution of the feasibility issue. Importantly, it demonstrates that under certain technical conditions, such as appropriate choice of loss functions and data sets, an optimal procedure for transfer learning exists.

However, I think the application of this framework is restrictive and the advantage of transfer learning (compared with learning from scratch) is omitted.

**Strengths:**

* This paper proposed a unified framework for transfer learning.
* They proposed several conditions, under which the transfer learning problem is feasible.

**Weaknesses:**

* There are several important settings this framework can't include. For example, the shared representation setting [1] and covariate shift [2]. These settings are commonly studied in both theoretical and algorithmic studies. So I think the application of this paper is restrictive.
* Besides, a more important problem is that I can't find evidence from this framework that transfer learning has some superiority compared with learning from scratch. Is learning a transformation between $X_S$ and $X_T$ or between $Y_S$ and $Y_T$ easier than directly learning the relationship between $X_T$ and $Y_T$? I think this is the central wisdom in transfer learning, which is omitted in this paper.


[1] Tripuraneni, Nilesh, Michael Jordan, and Chi Jin. "On the theory of transfer learning: The importance of task diversity." *Advances in neural information processing systems* 33 (2020): 7852-7862.

[2] Pathak, Reese, Cong Ma, and Martin Wainwright. "A new similarity measure for covariate shift with applications to nonparametric regression." *International Conference on Machine Learning*. PMLR, 2022.

**Questions:**

Please see Cons.

---

### Meta-Review · Area_Chair_1jD6 · 2023-12-19

**Metareview:**

The paper formulates a three-step optimization framework for transfer learning allowing for the resolution of the feasibility issue, that is, the framework allows to define certain technical conditions for the existence of an optimal solution.

The proposed framework is novel and interesting however the current version of the paper does not make clear enough how the results relate to previous theoretical results on transfer learning and what which novel insights can be gained. Making this more clear, would significantly strengthen the paper.

**Justification For Why Not Higher Score:**

All 4 reviewers voted for rejection. Its not clear enough what insights are provided by the framework and how it relates to previous theoretical results.

**Justification For Why Not Lower Score:**

N/A

---

### Decision · Program_Chairs · 2024-01-16

Reject